# The Comprehensive Analysis of Motor and Neuropsychiatric Symptoms in Patients with Huntington’s Disease from China: A Cross-Sectional Study

**DOI:** 10.3390/jcm12010206

**Published:** 2022-12-27

**Authors:** Yangfan Cheng, Xiaojing Gu, Kuncheng Liu, Tianmi Yang, Yi Xiao, Qirui Jiang, Jingxuan Huang, Junyu Lin, Qianqian Wei, Ruwei Ou, Yanbing Hou, Lingyu Zhang, Chunyu Li, Jean-Marc Burgunder, Huifang Shang

**Affiliations:** 1Department of Neurology, Rare Disease Center, West China Hospital, Sichuan University, Chengdu 610041, China; 2Laboratory of Neurodegenerative Disorders, National Clinical Research Center for Geriatric, West China Hospital, Sichuan University, Chengdu 610041, China; 3Department of Neurology, Swiss Huntington’s Disease Centre, Siloah, University of Bern, 3006 Bern, Switzerland

**Keywords:** Huntington’s disease, Chinese population, motor assessment, neuropsychiatric symptoms

## Abstract

Huntington’s disease (HD) is an autosomal dominant inherited neurodegenerative disorder caused by CAG repeats expansion. There is a paucity of comprehensive clinical analysis in Chinese HD patients due to the low prevalence of HD in Asia. We aimed to comprehensively describe the motor, neuropsychiatric symptoms, and functional assessment in patients with HD from China. A total of 205 HD patients were assessed by the Unified Huntington’s Disease Rating Scale (UHDRS), the short version of Problem-Behavior Assessment (PBA-s), Hamilton Depression Scale (HAMD) and Beck Depression Inventory (BDI). Multivariate logistic regression analysis was used to explore the independent variables correlated with neuropsychiatric subscales. The mean age of motor symptom onset was 41.8 ± 10.0 years old with a diagnostic delay of 4.3 ± 3.8 years and a median CAG repeats of 44. The patients with a positive family history had a younger onset and larger CAG expansion than the patients without a family history (*p* < 0.05). There was a significant increase in total motor score across disease stages (*p* < 0.0001). Depression (51%) was the most common neuropsychiatric symptom at all stages, whereas moderate to severe apathy commonly occurred in advanced HD stages. We found lower functional capacity and higher HAMD were independently correlated with irritability; higher HAMD and higher BDI were independently correlated with affect; male sex and higher HAMD were independently correlated with apathy. In summary, comprehensive clinical profile analysis of Chinese HD patients showed not only chorea-like movement, but psychiatric symptoms were outstanding problems and need to be detected early. Our study provides the basis to guide clinical practice, especially in practical diagnostic and management processes.

## 1. Introduction

Huntington’s Disease (HD) is an autosomal dominant hereditary neurodegenerative disease characterized by a triad of progressive motor abnormalities, psychiatric symptoms, and cognitive deterioration [1]. HD genetically is caused by an abnormal expansion of CAG repeats in the *IT15* gene on chromosome 4, which encodes an expanded polyglutamine stretch in the huntingtin protein [2]. The penetrance of HD is age-dependent, with a nearly full penetrance of CAG repeats over 40 at the age of 65 [3]. HD prevalence varies greatly among different ethnicities, with values of 4–10 per 100,000 in Caucasian populations [4], and lower with 0.1–1 per 100,000 in Asian populations [5,6,7,8]. It has been suggested that the much lower prevalence of HD in the Asian population might be due to genetic discrepancy including CAG repeats number, huntingtin haplotype, and CCG polymorphisms in the *IT15* gene [9]. Previously published studies on Chinese HD cases showed that clinical differences may be related to the different ethnic backgrounds between Chinese and European [10,11]. HD typically has a mid-adult onset and a disease duration of 15–20 years [12]. No disease-modifying treatment is available hitherto.

Based on the large multi-center HD cohorts established in populations of European ancestries, such as Track HD [13], Predict HD [14] and Enroll HD (www.enroll-hd.org), clinical characteristics of HD including demographic features, motor symptoms, psychiatric features, and cognitive performance have been well studied in European ancestry populations [12,15,16]. However, owing to the low prevalence of HD in Asia and underestimated patients, only a few systematic clinical studies on the Chinese HD population are available. The reported prevalence of psychiatric symptoms and disorders varies considerably due to diverse study populations at different stages of HD and the use of different assessment methods with varying definitions of psychiatric phenomena [17,18,19,20]. Our previous study with the limited number of patients (58 subjects) did not include non-motor symptoms analysis [21] and another study focused on genetic features and the majority of their patients were from southeastern China [22]. Further, there is a paucity of a comprehensive analysis of the clinical phenotype description of Chinese HD patients.

Therefore, in the current study, we have comprehensively (1) described the genetic profile and clinical characteristics including motor, psychiatric symptoms, and functional assessment as well in Chinese HD patients; (2) assessed the occurrence of neuropsychiatric symptoms, including affect, irritability, apathy, psychosis and dysexecutive behavior, and explored the correlates of common neuropsychiatric symptom in Chinese HD patients.

## 2. Method

### 2.1. Participants

HD subjects were recruited from the Department of Neurology of West China Hospital and their diagnosis were made by at least two senior neurologists according to patients’ clinical manifestations and positive genetic tests. The study was approved by the Ethics Committee of West China Hospital (approval number 2015-236). All participants signed an informed consent. All the clinical tests and neurological examinations were administered by experienced and specifically trained neurologists.

### 2.2. Assessment of Motor and Functional Capacity

Demographic information including sex, age, ethnicity, education, family history, and clinical information including age of onset were collected for each subject. Symptom onset was defined by family members’ or patients’ recollection of the initial clinical features. The age at onset is defined by the age when motor signs or symptoms occur. Juvenile-onset HD is defined patients with symptom onset before the age of 20 years and elder-onset HD is defined with onset older than the age of 60 [22]. Motor signs and functional capacity were assessed by the Unified Huntington’s Disease Rating Scale (UHDRS) [23]. The motor assessments were recorded as total motor score, which was computed by summing up 15 items. According to previous studies, the 15 items of the UHDRS motor scale can be grouped into seven subdomains, including eye movement (ocular pursuit, saccade initiation and saccade velocity), oropharyngeal (dysarthria and tongue protrusion), hand movements (finger taps, pronate/supinate-hands and Luria), rigidity/bradykinesia (rigidity-arms and bradykinesia-body), dystonia (dystonia), chorea (chorea) and gait/balance (gait, tandem walking and retropulsion pull test) [24]. Functional assessments included total functional capacity scale (TFC), functional assessment and independence assessment. The subjects were divided into five stages in the current study based on their TFC scores: stage 1 = TFC of 11–13; stage 2 = TFC of 7–10; stage 3 = TFC of 3–6; stage 4 = TFC of 1–2 and stage 5 =TFC of 0 score [25]. The UHDRS scale was used under license from the Chinese and European Huntington Study Group.

### 2.3. Assessment of Neuropsychiatric Problems and General Cognition

Neuropsychiatric problems were assessed by Hamilton Depression Scale (HAMD) [26], Hamilton anxiety scale (HAMA) [27], Beck Depression Inventory (BDI) [28], and the short version of Problem-Behavior Assessment (PBA-s) [29]. The PBA-s measured severity and frequency of 11 sub-items, according to previous factor analyses, which could be grouped into five subscales, which are irritability, apathy, affect, dysexecutive and psychosis [29,30]. The irritability subscale (range 0–32 points) consisted of two items: “irritability” and “aggression”. Participants were divided into three groups, depending on their total subscale score: no symptoms (≤2 points), mild symptoms (3–8 points), or moderate to severe symptoms (>8 points). The apathy subscale (range 0–32 points) consisted of two items: “apathy” and “perseveration”. Participants were divided into three groups, depending on their total subscale score: no symptoms (≤2 points), mild symptoms (3–8 points), or moderate to severe symptoms (>8 points). The affect subscale (range 0–48 points) consisted of three items: “depressed mood”, “anxiety”, and “suicidal ideation”. Participants were divided into three groups, depending on their total subscale score: no symptoms (≤3 points), mild symptoms (4–12 points), or moderate to severe symptoms (>12 points). The dysexecutive subscale (range 0–16 points) consisted of one item: “behavior suggestive of disorientation”. Participants were divided into three groups, depending on their total subscale score: no symptoms (≤1 points), mild symptoms (2–3 points), or moderate to severe symptoms (>4 points). The psychosis subscale (range 0–48 points) consisted of three items: “obsessive compulsive behavior”, “delusion”, and “hallucination”. Participants were divided into three groups, depending on their total subscale score: no symptoms (≤3 points), mild symptoms (4–12 points), or moderate to severe symptoms (>12 points). A sub-score of each item was computed by multiplying the severity score and frequency score of that item. Total scores for each subscale were computed by summing the total scores of the separate items of each subscale. The score on the Mini-Mental State Examination was used for screen general cognitive abilities (range 0–30 points). The score after education correction below 27 points was defined cognitive decline. The PBA-s scale was used under license from the Chinese and European Huntington Study Group.

### 2.4. Statistical Analysis

Data are presented as *n* (%), mean (± standard deviation, SD) if the data distributed normally. The normality was tested by Kolmogorov–Smirnov test and Shapiro–Wilk test. For comparison between two continuous variables, *t*-tests, Wilcoxon tests, or Mann–Whitney were used as appropriate. ANOVA was used for comparison between multi-group. Correlation was calculated by Pearson or Fisher exact Chi-Square when appropriate. The independent correlates of each subscale of neuropsychiatric problems were determined by comparing HD patients without subscale symptoms with HD patients with mild, moderate to severe subscale symptoms using multivariate logistic regression analysis. Variables with a *p* value ≤ 0.05 in the univariate analyses were included in these multivariate analyses. Due to the low prevalence of the psychosis and dysexecutive subscale, we did not perform the multivariate analyses on these two subscales.

Statistical analysis was conducted using Prism GraphPad (GraphPad Software, San Diego, CA, USA) and Statistic Package for Social Science (SPSS) version 22.0 (IBM Corp., Armonk, NY, USA). Multiple comparisons were adjusted with Bonferroni correction. And *p* < 0.05 or *p* < 0.05/*n* (Bonferroni correction) was considered statistically significant. 

## 3. Results

### 3.1. Demographic and Genetic Data

Our study enrolled 205 genetically confirmed HD patients, including 84 males and 121 females. Demographic and clinical characteristics of the 205 HD patients are presented in Table 1. The majority of the patients were from the southwest of China. The mean age of all patients was 52.9 ± 11.5 years, and their estimated median disease duration was 4.5 years (IQR: 2.3–6.9). Of all patients, 162 had clear transmission mode (86 paternal and 76 maternal inheritance) and 43 denied family members had any known chorea-like movements. Of the total patients, 188 patients presented with motor signs initially (91.7%, 76 males and 112 females), 5 patients presented with cognitive decline initially (2.4%, 3 males and 2 females) and 12 patients showed psychiatric symptoms at the first diagnosis and developed motor symptom later (5.8%, 5 males and 7 females).

The mean age of motor symptom onset was 41.8 ± 10.0 (range 17–69) years old, and among them, the group of 40–45 years accounted for the highest proportion (21.0%) (Figure 1a). According to motor symptom onset, four (2.0%) had a juvenile-onset with a mean age of 18.1 ± 1.4 (range 17–20) years old, and eight (3.9%) had an elder age onset of 63.5 ± 3.1 (range 61–69) years. The patients with a positive family history had a significantly younger motor symptom onset (*p* < 0.001) than patients with a negative family history. The patients who initially developed motor, cognitive and psychiatric symptoms respectively had a different trend of onset (*p* = 0.076). Among them, HD patients with cognitive decline initial presentation had an older trend of onset than those with motor symptoms initial presentation (*p* = 0.062).

The median CAG repeats numbers of all patients were 44 (IQR: 42–48). Motor symptom onset was inversely correlated with the number of CAG repeats (*p* < 0.001) and the number of CAG repeats accounted for 42.6% of the variance in motor symptom onset (*r* = −0.65, R^2^ = 0.426) (Figure 1b). Patients with a positive family history had a larger number of CAG expansion than the patients without a family history (*p* = 0.007). The number of CAG repeat of patients with initial presentation of cognitive decline was less than that of patients with initial presentation of motor onset (*p* = 0.017).

The mean age at diagnosis of all patients was 45.7 ± 11.1 years old with 4.3 ± 3.8 years diagnostic delay. The age at diagnosis of patients with a positive family history was younger than that of patients without a family history (*p* < 0.001). Similarly, the diagnostic delay of patients with a positive family history was shorter than patients without a family history (*p* = 0.002). The disease duration was longer in patients without a positive family history than patients with a positive family history (*p* = 0.006).

### 3.2. Motor Symptoms and Functional Capacity

The mean total motor score of all patients was 40.5 ± 19.0. Motor signs assessed by UHDRS are shown in Table 2. Elderly onset HD patients had the highest mean motor score followed by adult-onset HD and juvenile HD patients (*p* = 0.027). A significantly higher score of the chorea subdomain was found regarding the age of disease onset. The highest mean score of the chorea subdomain was observed in elderly onset HD patients, while, the lowest score was observed in the juvenile HD patients (*p* = 0.005) (Appendix A). Male patients showed potentially more severe motor symptoms than females, especially in subdomains of dystonia (*p* = 0.015). No differences were found in eye movement, oropharyngeal part, hand movement, rigidity/bradykinesia, dystonia, and gait/balance grouped by initial symptom, age of onset, and family history.

Patients with a positive family history had higher score of functional assessment scores (FAS) than patients without a family history (*p* = 0.002). In addition, patients with a positive family history were more independent than patients without a family history (*p* = 0.045). The highest mean score of independence assessment was observed in the juvenile-onset HD patients, and the lowest in the elderly onset HD patients (*p* = 0.016). However, no difference was found in the total motor score, FAS, independence assessment, and TFC grouped by different initial symptoms. 

In addition, we found that there was a significantly increased motor score across each stage (*p* < 0.0001) (Figure 2a). Among 7 subdomains, eye movement, oropharyngeal, hand movement, chorea, and gait/balance deteriorated with stage, while rigidity/bradykinesia and dystonia only worsened in the late stage (Table 3 and Figure 2b).

### 3.3. Neuropsychiatric Symptoms 

Of all patients, affect (reported in 93/182, 51.1%) was the most common neuropsychiatric symptom domain. Apathy (reported in 77/183, 42.1%) and irritability (reported in 74/183, 40.4%) were the second and the third most common neuropsychiatric symptom domains. In addition, the frequency of moderate to severe non-motor symptoms at baseline was 18.0% apathy, 14.2% irritability, 11.5% affect, 4.4% dysexecutive behavior and 1.1% psychosis (Figure 3a). Of patients who were in stage 1, the most common moderate to severe neuropsychiatric symptom was irritability (10%); in stage 2, the most common moderate to severe neuropsychiatric symptom was affect (14%) and in advanced HD patients who were in stage 3–5, the most common moderate to severe neuropsychiatric symptom was apathy (42%) (Figure 3b).

Univariate analyses showed that HD patients with affect problems had higher HAMD scores (*p* = 0.002) and higher BDI scores (*p* = 0.017). Higher scores of HAMD (*p* = 0.016) and BDI (*p* = 0.035) were independently correlated with the affect domain. Univariate analyses showed that HD patients with irritability problems had a positive family history (*p* = 0.049), a longer diagnostic delay (*p* = 0.003), a higher total motor score (*p* < 0.001), a lower TFC (*p* = 0.001), lower MMSE score (*p* = 0.016), higher HAMD score (*p* < 0.001), and BDI score (*p* = 0.004). Lower TFC (*p* = 0.03) and higher score of HAMD (*p* = 0.002) were independently correlated with the irritability domain. Univariate analyses showed that HD patients with moderate to severe apathy were more male (*p* = 0.04), had a longer diagnostic delay (*p* = 0.002), a higher total motor score (*p* < 0.001), a lower TFC (*p* < 0.001), lower MMSE score (*p* = 0.019), higher HAMD score (*p* < 0.001) and BDI score (*p* = 0.005). Male sex (*p* = 0.03) and higher score of HAMD (*p* = 0.007) were independently correlated with the apathy domain (Table 4). We did not perform the univariate and multivariate analyses of psychosis and dysexecutive domains due to their low prevalence.

Moreover, no statistical differences were found in the score of HAMD, HAMA, and BDI regarding different initial symptoms and motor AAO. Patients without a family history had higher scores of HAMA than patients with a positive family history (*p* = 0.016). The score of neuropsychiatric and cognitive scales including HAMD, HAMA, BDI, and MMSE of all patients are shown in Appendix A.

## 4. Discussion

We provide comprehensive documentation and analysis of the clinical characteristics including motor and psychiatric symptoms as well as the genetic profile based on a larger population of southwestern Chinese HD patients. This is to get a solid foundation for HD clinical, genetic, and therapeutic research in China.

The median CAG repeats number of our patients was similar to that of Korean [31] (45.4 ± 4.7), South-African [32] (46 ± 3), Dutch [33] (46 ± 3), and Thailand cohorts [34] (43.5 ± 3), while less than that of Mexican HD patients [35] (47.2 ± 5.4). The age of motor symptom onset was similar to the 40.3 ± 11.9 years old reported by Li, H.L. [22]; however, elder than that of Mexicans [35] (37.4 ± 12.9 years) and Thailand HD patients [34] (37.3 ± 8.3 years), earlier than that of Korean [31] (46.5 ± 12.7 years) and Dutch HD patients [33] (47.0 ± 15.0 years). In our cohort, only 42.6% of the variation for motor symptom onset could be explained by the CAG repeats number, confirming the previous results [11]. This suggests other genetic and environmental factors contribute more importantly to the remaining variation of disease onset [36]. The Li, H.L.’s study had similar demographic and genetic features to our study, while we made more comprehensive analyses of clinical characteristics including motor and neuropsychiatric symptoms [22] (Appendix A).

Over 90% of HD patients developed initial presentation of motor symptoms such as chorea-like abnormal movement in our cohort, which was similar to the Korean population (32/36, 89%) [31], while greater than American (352/510, 69%) [37] and Mexican HD patients (366/691, 53%) [35]. Our study found that HD patients who developed an initial presentation of cognitive decline had a lower number of CAG repeats than those who developed an initial presentation of motor symptoms, which was inconsistent with the previous study conducted by Li, H.L. [22]. The differences in the sample size, genetic background, and environmental modification among different ethnic populations and regions may contribute to the such discrepancy. 

We noticed that the patients without positive family history had an older age at diagnosis and longer diagnostic delay (more than 5 years) than patients with family history, which suggests the diagnosis of HD patients without family history is delayed. Therefore, for patients with chorea-like movement, cognitive decline, or psychiatric symptoms but without a family history, physicians need to use genetic testing to confirm the diagnosis of HD.

Elderly onset HD patients were suggested more support from family, leading to more burden of care because of more severe motor symptoms and less independence. Consistent with the previous finding, juvenile-onset HD patients presented fewer chorea movements and showed big heterogeneity including epilepsy [38,39], gait instability [22], intellectual decline [40], and parkinsonism [40,41] in juvenile-onset HD patients. Moreover, patients with a positive family history were more independent and had a higher functional score than the patients without a family history, which might relate to the younger disease onset and shorter diagnostic delay of patients with a positive family history. Hence, it is important to identify patients in the initial phase, and early use of medication to control motor symptoms could improve patients’ quality of life. 

The total motor score was increased with the disease stage identified in our patients, which was consistent with previous cross-sectional and longitudinal studies on European ancestry HD patients [16,42,43,44]. By Fingin’s classification, the total eye movement score of our patients was increased with disease stage, which was in accordance with the previous studies, indicating a progressive neurodegeneration of the frontal lobe [11,45]. The score of rigidity/bradykinesia and dystonia of our patients was increased in the late stage but not in the early stage, which was consistent with some previous studies [42,44,46]. Therefore, in the late stage of HD, the management of features of parkinsonism and dystonia should pay attention.

This is the first systematic analysis of neuropsychiatric disorders of Chinese HD patients. We found neuropsychiatric symptoms are very common in HD patients in all stages, especially affect, apathy, and irritability. Only 23% of HD patients did not report any neuropsychiatric symptoms on the first visit to our center. The prevalence of psychosis and dysexecutive function in our HD population with a median duration of disease of 5.2 years was low. A previous study reported that depressed mood and sadness were early symptoms of HD [47], while significantly lower rates of depression were present in the advanced stages of the disease [48,49]. Our study found the prevalence of affect in HD patients increased as the disease progressed, while the score of TFC did not correlate with the severity of affect independently, which was inconsistent with the previous study [50]. Considering the common symptom of moderate to severe apathy in advanced HD patients, the assessment of affect might be influenced by the comorbidity of apathy. We found that the score of HAMD was independently inversely correlated with the severity of affect, irritability, and apathy, which suggests HAMD had a sensitive diagnostic value in identifying neuropsychiatric symptoms in HD patients, not only depression. Moreover, it suggests neuropsychiatric symptoms are often present in a co-morbid form in patients with advanced HD. In addition, the TFC score was independently inversely correlated with the severity of irritability, suggesting that severe irritability is more prevalent in advanced disease stages and may be the only neuropsychiatric symptom that is linearly related to progressive neurodegeneration. Whereas previous studies reported that apathy increased with HD progression [21,51,52], which was inconsistent with our findings. Further investigations are needed to solve this inconsistency. Therefore, neuropsychiatric symptoms in HD patients are challenging and needed to be focused on. Moderate to severe neuropsychiatric symptoms could severely affect the quality of life of both patients and caregivers. Considering the high prevalence of apathy in advanced stage, the use of antipsychotics and benzodiazepines needs to be cautious. An antidepressant is the preferred pharmacologic option when there is difficulty differentiating depression from apathy in HD patients [53].

Although our study provided valuable results, it did have some drawbacks. First, because of the delay in diagnosis, the onset symptom of patients was sometimes blurry. There might be some recalling bias by a retrospective inquiry from the family members. Second, we did not consider the intervention effect of medication when analyzing the neuropsychiatric symptoms despite the very low rate of antidepressant drugs at the first visit. Third, because this was a single-center study, there might be some population bias. A multicenter study is needed to validate our findings.

In summary, we summarized the comprehensive clinical and genetic profiles of Chinese HD patients. In China, the delay from disease onset to diagnosis is over 4 years, especially for patients without a family history, which suggests neurologists should also pay more attention to the diagnosis of HD when patients who presented chorea-like movement, cognitive decline, or psychiatric symptoms without a positive family history. The genetic test needs to be carried out to confirm the diagnosis. The neuropsychiatric problems of Chinese HD patients were relatively common across all stages of the disease. These symptoms lead to important impairments in the quality of life, and place a significant burden on healthcare services. Therefore, it is important for physicians to specifically check these symptoms for early detection and more successful management. These data are of great significance in guiding clinical practice, especially in practical diagnostic processes and medication management. Further longitudinal following up is needed to investigate the whole natural history of the disease in China.

## Figures and Tables

**Figure 1 jcm-12-00206-f001:**
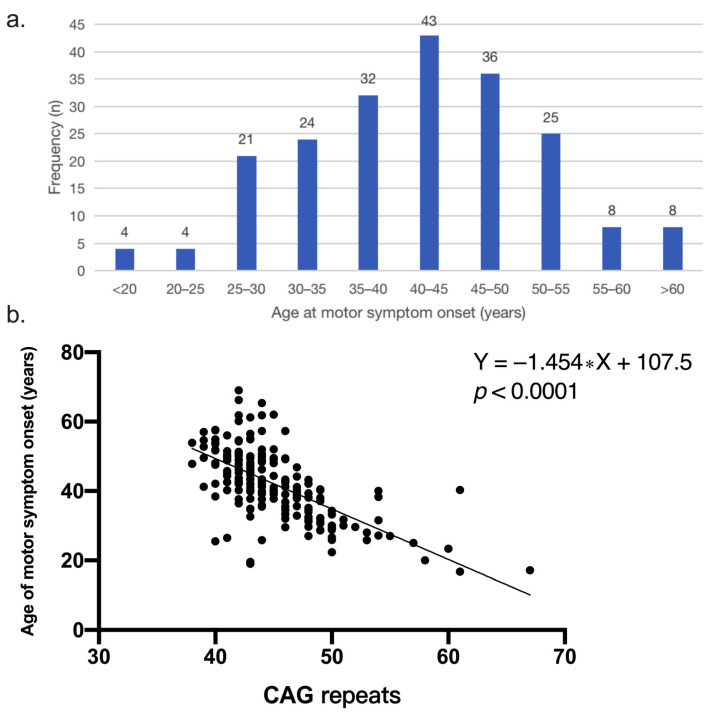
(**a**) Frequency distribution of age at motor symptom onset in Chinese HD patients (*n* = 205). The group of 40–45 years accounted for the highest proportion. (**b**) Correlation between CAG repeats and age of onset (*p* < 0.0001). The simple linear regression equation is Y = −1.454 ∗ X + 107.5.

**Figure 2 jcm-12-00206-f002:**
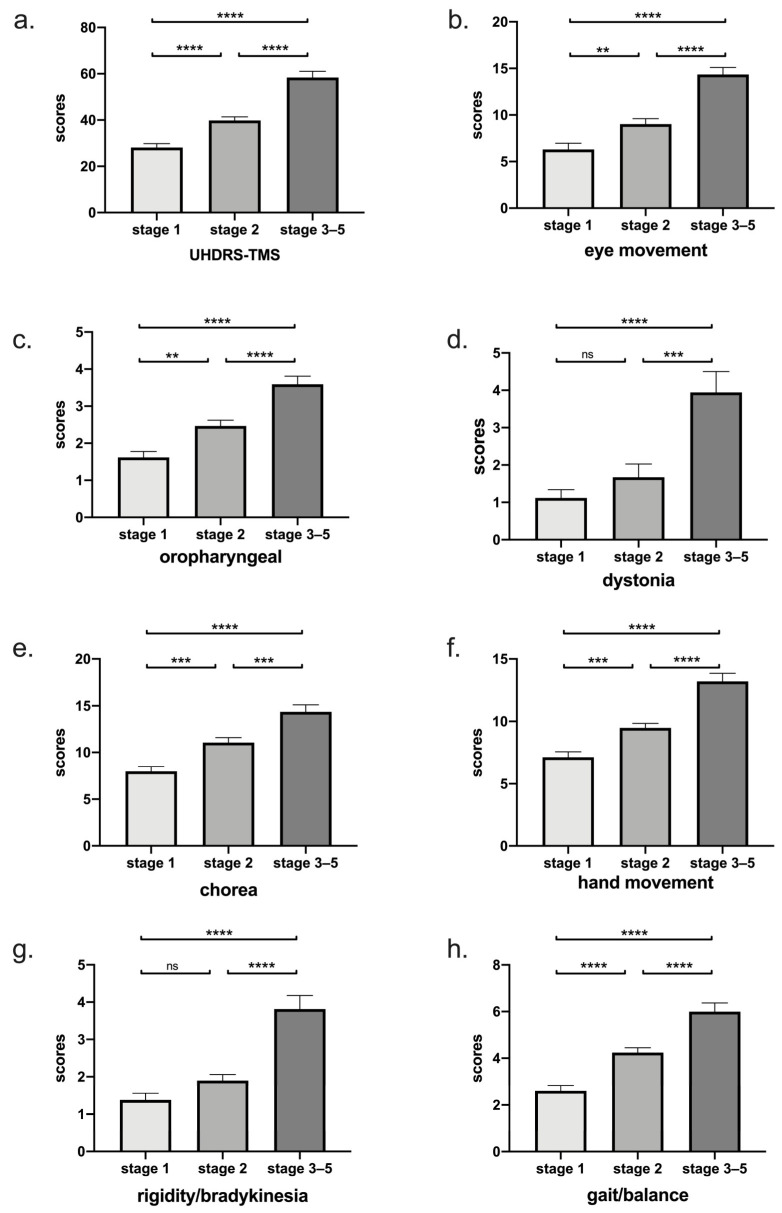
(**a**) Total motor score grouped by disease stage was shown. There was a significant increase in total motor score across each stage (*p* < 0.0001). (**b**–**h**) According to Fingin’s classification, scores of seven subdomains were shown. Eye movement, oropharyngeal, hand movement, chorea, and gait/balance deteriorated with stage, while rigidity/bradykinesia and dystonia only worsened in the late stage. ****: *p* < 0.0001; ***: *p* < 0.001; **: *p* < 0.01; ns: *p* > 0.05.

**Figure 3 jcm-12-00206-f003:**
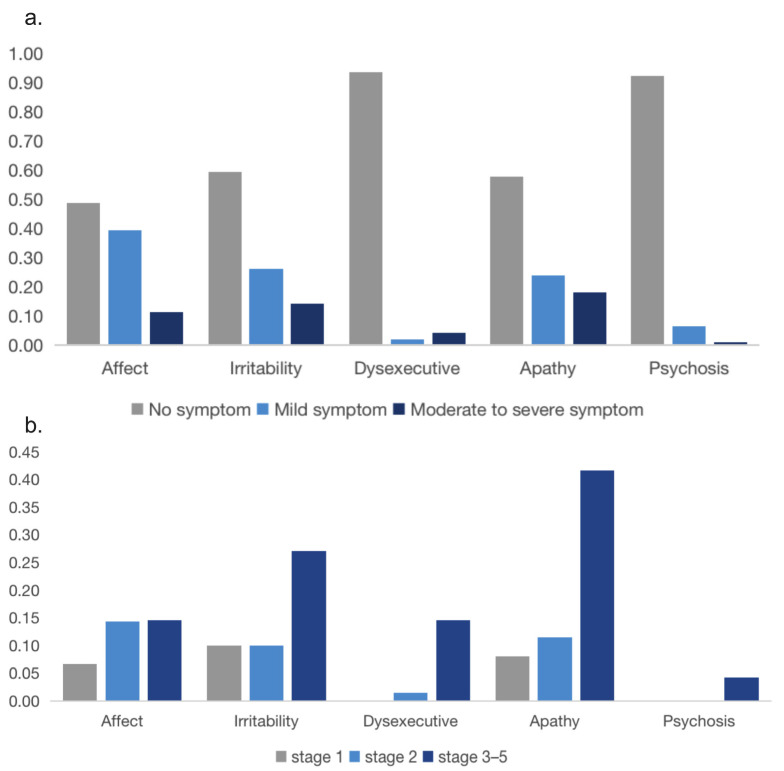
(**a**) Prevalence of neuropsychiatric symptoms in HD patients at first diagnosis. (**b**) Prevalence of moderate to severe symptoms on the behavioral subscales across different disease stages.

**Table 1 jcm-12-00206-t001:** Demographic and clinical characteristics of the 205 HD patients.

	Age of Motor Symptom Onset (Years)		CAG Repeats		Age at Diagnosis (Years)		DD (Years)		Duration (Years)	
	Mean (SD)	Range	Mean (SD)	Range	Mean (SD)	Range	Mean (SD)	Range	Mean (SD)	Range
Total (*n* = 205)	41.8 (10.0)	17–69	45.2 (4.5)	38–67	45.7 (11.1)	18–78	4.3 (3.8)	0–20	5.2 (3.9)	0–20
Male (*n* = 84)	42.1 (10.5)	18–67	45.2 (4.7)	38–67	46.6 (11.6)	22–78	4.7 (4.1)	0–18	5.5 (4.1)	0–18
Female (*n* = 121)	41.6 (9.6)	17–69	45.2 (4.4)	38–61	45.2 (10.7)	18–74	4.0 (3.6)	0–20	5.1 (3.7)	0–20
Juvenile HD (*n* = 4)	18.1 (1.40)	17–20	53.5 (12.40)	43–67	22.10 (3.1)	18–25	4.0 (2.0)	1–5	4.50 (1.0)	3–5
Adult HD (*n* = 193)	41.4 (8.6)	20–58	45.1 (4.2)	38–61	45.3 (9.80)	19–70	4.1 (3.6)	0–19	5.2 (3.7)	0–19
Elderly-onset HD (*n* = 8)	63.5 (3.1)	61–69	43.0 (1.2)	42–45	68.2 (5.4)	62–78	7.4 (7.3)	1–20	7.4 (7.3)	1–20
Positive family history (*n* = 162)	40.1 (9.4) ^a^	17–67	45.7 (4.6) ^b^	38–67	43.6 (10.1) ^d^	18–69	3.8 (3.4) ^e^	0–19	4.8 (3.5) ^f^	0–19
Paternal inheritance (*n* = 86)	38.9 (9.7)	17–67	46.4 (5.2)	39–67	42.2 (10.2)	18–69	3.3 (2.8)	0–12	4.38 (2.9)	0–15
Maternal inheritance (*n* = 76)	41.4 (9.0)	19–62	44.9 (3.8)	38–57	45.1 (9.8)	19–69	4.4 (4.0)	0–19	5.2 (4.0)	0–19
Negative family history (*n* = 43)	48.4 (9.4) ^a^	28–69	43.2 (3.3) ^b^	38–53	53.9 (10.7) ^d^	31–78	5.8 (4.8) ^e^	0–20	7.0 (4.8) ^f^	1–20
Motor (*n* = 188)	41.4 (9.7)	17–67	45.2 (4.5) ^c^	38–67	45.5 (10.8)	18–78	4.2 (3.7)	0–19	5.2 (3.8)	0–19
Cognitive (*n* = 5)	49.6 (7.5)	42–62	43.4 (1.1) ^c^	42–45	50.1 (7.2)	42–62	3.8 (1.4)	2–6	3.7 (1.4)	2–6
Psychiatric (*n* = 12)	45.6 (13.5)	25–69	45.8 (5.8)	40–57	47.8 (15.3)	19–73	5.9 (6.0)	0–20	6.9 (5.4)	0–20

a–f: statistically significant was found between the groups (*p* < 0.05). DD: diagnostic delay; SD: standard deviation.

**Table 2 jcm-12-00206-t002:** Motor measures by the Unified Huntington’s Disease Rating Scale (UHDRS) scores of 205 HD patients.

	UHDRS I (Total Motor Score)	UHDRS II (FAS)	UHDRS III (Independence Assessment)	UHDRS IV (TFC)
Total (205)	40.5 (19.0)	18.3 (5.4)	82.9 (15.8)	8.6 (3.4)
Male	43.2 (20.1)	18.5 (5.5)	82.2 (16.5)	8.6 (3.4)
Female	38.5 (18.0)	18.2 (5.4)	83.4 (15.3)	8.6 (3.4)
Juvenile HD (4)	22.0 (1.4) ^a^	17.5 (7.1)	93.3(5.8) ^c^	11.3 (0.6)
Adult HD (193)	40.4 (18.8) ^a^	18.5 (5.2)	84.1 (13.8) ^c^	8.6 (3.4)
Elderly-onset HD (8)	54.4 (11.4) ^a^	14.1 (8.0)	70.6 (21.3) ^c^	6.8 (4.2)
Positive family history (162)	40.3 (19.0)	19.4 (4.2) ^b^	84.0 (15.2) ^d^	8.8 (3.3)
Paternal inheritance (86)	41.0 (18.4)	19.2 (4.3)	84.6 (12.4)	8.7 (3.1)
Maternal inheritance (76)	40.4 (21.3)	19.4 (4.4)	83.2 (17.8)	8.8 (3.5)
Negative family history (43)	41.2 (19.1)	16.3 (6.1) ^b^	78.9 (17.4) ^d^	7.8 (3.7)
Motor (188)	40.4 (19.2)	18.4 (5.4)	83.2 (15.6)	8.6 (3.4)
Cognitive (5)	43.2 (24.0)	17.3 (3.2)	74.0 (27.0)	6.6 (3.1)
Psychiatric (12)	41.1 (14.2)	18.6 (3.0)	82.7 (12.3)	8.3 (2.5)

a–d: statistically significant was found among the groups (*p* < 0.05). UHDRS: Unified Huntington’s Disease Rating Scale; FAS: functional assessment scores; TFC: total functional capacity.

**Table 3 jcm-12-00206-t003:** Motor assessment across stages and grouped by Fingin’s classification.

Sub-Items	Total (*n* = 205)	Stage 1 (*n* = 68)	Stage 2 (*n* = 79)	Stages 3, 4, and 5 (*n* = 54)	*p*	Stage 1 vs. 2	Stage 2 vs. 3 and 4	Stage 1 vs. 3 and 4
Total motor score	40.8 (19.6)	28.1 (14.1)	39.8 (13.8)	58.4 (20.1)	<0.0001	<0.0001	<0.0001	<0.0001
Subdomain								
Eye movement subscale	9.2 (6.5)	6.3 (5.6)	9.0 (5.2)	13.4 (6.7)	<0.0001	0.013	<0.0001	<0.0001
Oropharyngeal subscale	2.4 (1.6)	1.6 (1.3)	2.5 (1.4)	3.6 (1.6)	<0.0001	0.001	<0.0001	<0.0001
Hand movement subscale	9.5 (4.6)	7.1 (3.6)	9.5 (3.3)	13.2 (4.7)	<0.0001	0.0002	<0.0001	<0.0001
Rigidity/bradykinesia subscale	2.2 (2.1)	1.4 (1.5)	1.9 (1.4)	3.8 (2.7)	<0.0001	0.098	<0.0001	<0.0001
Dystonia subscale	2.1 (3.3)	1.1 (1.8)	1.7 (3.2)	3.9 (4.1)	<0.0001	0.467	0.003	<0.0001
Chorea subscale	10.7 (5.5)	8.0 (4.0)	11.1 (4.8)	14.4 (5.5)	<0.0001	0.0004	0.0004	<0.0001
Gait/balance	4.2 (2.5)	2.6 (1.9)	4.2 (1.8)	6.0 (2.7)	<0.0001	<0.0001	<0.0001	<0.0001

**Table 4 jcm-12-00206-t004:** Characteristics and correlates of neuropsychiatric symptoms including affect, irritability, and apathy in HD mutation carriers.

	Affect						Irritability						Apathy					
	Univariate Analyses				Multivariate Logistic Regression		Univariate Analyses				Multivariate Logistic Regression		Univariate Analyses				Multivariate Logistic Regression	
	No Symptom	Mild	Moderate to Severe	*p* Value	OR (95% CI)	*p* Value	No Symptom	Mild	Moderate to Severe	*p* Value	OR (95% CI)	*p* Value	No Symptom	Mild	Moderate to Severe	*p* Value	OR (95% CI)	*p* Value
Male (%)	32 (32.3%)	26 (36.1%)	5 (23.8%)	0.751	/	/	45 (41.2%)	15 (31.2%)	12 (46.1%)	0.671	/	/	35 (33%)	21 (47.7%)	16 (48.4%)	0.041	0.31 (0.11–0.90)	0.032
Positive family history	71 (79.8%)	57 (79.2%)	15 (71.5%)	0.872	/	/	91 (83.5%)	35 (73%)	18 (69.3%)	0.049	/	/	86 (81.2%)	33 (75%)	25 (75.8%)	0.732	/	/
Age of onset (years)	41.20 (10.04)	41.48 (10.35)	41.52 (8.83)	0.981	/	/	41.92 (9.70)	42.86 (10.53)	41.33 (9.13)	0.782	/	/	41.34 (10.24)	42.59 (8.77)	43.77 (9.70)	0.431	/	/
CAG repeats, number	45 (43–48)	45 (43–48)	45 (42.5–48)	0.901	/	/	44 (43–47.5)	44 (42.25–48)	45 (41–47)	0.742	/	/	44 (42–48)	45 (43–48)	43 (42–46)	0.533	/	/
DD, years	4.17 (2.5–5.96)	4.17 (2.60–5.48)	3 (2–7.21)	0.871	/	/	3.2 (2.08–6.1)	4.38 (2.12–5.75)	6.34 (3.4–10.52)	0.003	1.02 (0.86–1.21)	0.80	3.29 (2.08–5.99)	4.38 (2.06–7.35)	6.09 (3.87–9.12)	0.002	1.06 (0.90–1.24)	0.492
Total motor score	40 (30–47)	39 (30.25–44.75)	41 (30.5–51)	0.633	/	/	33 (26–43.75)	40.5 (30.25–46)	50.5 (34.00–60)	<0.001	0.99 (0.94–1.03)	0.60	35.00 (24.5–43.5)	39 (29–43)	46 (36–55)	<0.001	0.96 (0.92–1.01)	0.132
TFC	8 (5–10)	8 (5–10)	8 (5–10)	0.854	/	/	10 (8–12)	8 (6–10)	6.5 (5–10.5)	0.001	0.82 (0.68–0.98)	0.03	10 (7–12)	9 (6.25–10)	5 (4–8)	<0.001	0.85 (0.68–1.07)	0.161
MMSE	24 (18–26)	24 (19.5–26)	23.5 (16.5–26.75)	0.901	/	/	25 (20–27)	11 (7–17)	23 (20.25–25.75)	0.016	1.03 (0.40–1.18)	0.61	24 (20.5–27)	24 (19.5–26)	22 (15–25)	0.019	1.02 (0.90–1.05)	0.801
HAMD	13.68 (6.01)	12.03 (5.06)	18.92 (6.37)	0.002	1.14 (1.03–1.28)	0.016	8.37 (5.82)	11.96 (6.58)	15.87 (6.79)	<0.001	1.15 (1.05–1.25)	0.002	7.30 (5.50)	12.52 (6.43)	15.50 (5.74)	<0.001	1.18 (1.05–1.33)	0.007
HAMA	11.44 (4.16)	10.67 (4.89)	15.00	0.672	/	/	8.63 (3.09)	10.75 (7.5)	13.00 (5.94)	0.332	/	/	9 (4.62)	12.50 (7.23)	12.00 (1.41)	0.481	/	/
BDI	9.16 (6.64)	7.63 (5.16)	14.50 (8.59)	0.017	1.14 (1.01–1.29)	0.035	4.87 (3.84)	8.37 (8.01)	11.00 (7.06)	0.004	1.00 (0.89–1.12)	0.96	4.84 (4.46)	9.09 (7.24)	8.91 (6.63)	0.005	1.00 (0.89–1.12)	0.962

DD: diagnostic delay; SD: standard deviation; UHDRS: Unified Huntington’s Disease Rating Scale; FAS: functional assessment scores; TFC: total functional capacity; MMSE: Mini-Mental State Examination; HAMD: Hamilton Depression Scale; HAMA: Hamilton anxiety scale; BDI: Beck Depression Inventory.

## Data Availability

Data and materials are available from the corresponding author on reasonable request.

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
