# Peer review of "The Comprehensive Analysis of Motor and Neuropsychiatric Symptoms in Patients with Huntington’s Disease from China: A Cross-Sectional Study"

_jcm, 2022, doi:10.3390/jcm12010206_

Round 1

Reviewer 1 Report

-Please check the guidelines for JCM abstract writing, it should be nonstructured

-The authors should remove acronyms such as TMS (total motor score), since TMS is the generally accepted method for neurophysiological studies investigating corticospinal excitability and refers to transcranial magnetic stimulation. Also, motor age of onset (AAO) is also an acronym that might be removed.

-The TFC is inserted into the aim of study?

-The list of tests, the full term, and acronyms were introduced in the Introduction section and should be consistently used in the rest of the manuscript without the need to repeat the whole term (paragraph 2.3). When introducing the title of the test, full term with acronym, and with the reference should be given (in the aim or in the methodology more acceptable). 

-Discussion please rewrite the first sentence of the first paragraph may be divided into two sentences.

-Since no raws were shown for review purposes, here is the sentence only to remove the term “always” and to delete it or insert the appropriate term.

“We noticed that the patients without positive family history had older age at diagnosis and longer diagnostic delay (more than 5 years) than patients with family history, which suggests the diagnosis of HD patients without family history is always delayed.”

-Is this sentence finished ? “Whereas previous studies reported that apathy increased with HD progression[20, 45].”

-This sentence need to be rewritten “In summary, we first summarized the comprehensive clinical and genetic profile of Chinese HD patients.” “We first”?

-The term “natural history” stands for?

-Please see some references related to the topic of your paper to include them if possible:

1

 Brain Sci. 2017 Jun; 7(6): 67.

Published online 2017 Jun 16. doi: 10.3390/brainsci7060067

 PMCID: PMC5483640

PMID: 28621715

 Neuropsychiatric Burden in Huntington’s Disease

Ricardo Augusto Paoli,1 Andrea Botturi,1,2,* Andrea Ciammola,3 Vincenzo Silani,3,4 Cecilia Prunas,1 Claudio Lucchiari,5 Elisa Zugno,1 and Elisabetta Caletti1

2

A Systematic Review of Neuropsychiatric Symptoms and Functional Capacity in Huntington’s Disease

Jessie Sellers, M.S.N., Sheila H. Ridner, Ph.D., R.N., Daniel O. Claassen, M.D., M.S.

 Published Online:30 Aug 2019https://doi.org/10.1176/appi.neuropsych.18120319

3

J Neuropsychiatry Clin Neurosci

 . 2022 Spring;34(2):100-112. doi: 10.1176/appi.neuropsych.21060154. Epub 2021 Dec 28.

Apathy and Huntington's Disease: A Literature Review Based on PRISMA

 Jihene Matmati 1, Christophe Verny 1, Philippe Allain 1

 PMID: 34961332 DOI: 10.1176/appi.neuropsych.21060154

4

Archives of Psychiatric Nursing

Volume 35, Issue 3, June 2021, Pages 284-289

Archives of Psychiatric Nursing

Lifetime neuropsychiatric symptoms in Huntington's disease: Implications for psychiatric nursing

Author links open overlay panelJessie S.Gibsona12David A.Isaacsa2Daniel O.ClaassenaJeffrey G.Stovallb

https://doi.org/10.1016/j.apnu.2021.03.006

5

J Neuropsychiatry Clin Neurosci

. 2022 Sep 21;appineuropsych21070191. doi: 10.1176/appi.neuropsych.21070191. Online ahead of print.

Apathy and Depression in Huntington's Disease: Distinct Longitudinal Trajectories and Clinical Correlates

Michael H Connors 1, Armando Teixeira-Pinto 1, Clement T Loy 1

PMID: 36128678 DOI: 10.1176/appi.neuropsych.21070191

6

Australas Psychiatry

. 2018 Aug;26(4):366-375. doi: 10.1177/1039856218791036. Epub 2018 Jul 16.

Huntington's disease: Neuropsychiatric manifestations of Huntington's disease

Anita My Goh 1, Pierre Wibawa 2, Samantha M Loi 3, Mark Walterfang 4, Dennis Velakoulis 3, Jeffrey Cl Looi 5

 PMID: 30012004 DOI: 10.1177/1039856218791036

Author Response

Response to Reviewer 1 Comments

Comment 1

-Please check the guidelines for JCM abstract writing, it should be nonstructured.

 Respond Thank you for pointing out our mistake. We have rewritten the abstract to meet the guidelines of JCM instructions. Please see the edited manuscript. I am willing to re-edit if needed.

Comment 2

-The authors should remove acronyms such as TMS (total motor score), since TMS is the generally accepted method for neurophysiological studies investigating corticospinal excitability and refers to transcranial magnetic stimulation. Also, motor age of onset (AAO) is also an acronym that might be removed.

 Respond Thanks for your explanations and helpful suggestions. We have removed acronyms of TMS and AAO and changed them into the appropriate full name. Please see the edited manuscript.

Comment 3

-The TFC is inserted into the aim of study?

 Respond Thanks for your helpful suggestions. We have added the functional assessment into the aim of study. Please see the abstract and the introduction part.

 Comment 4

-The list of tests, the full term, and acronyms were introduced in the Introduction section and should be consistently used in the rest of the manuscript without the need to repeat the whole term (paragraph 2.3). When introducing the title of the test, full term with acronym, and with the reference should be given (in the aim or in the methodology more acceptable).

 Respond Thanks for your helpful suggestion. We have double checked the acronyms were always consistently used in the manuscript and deleted the repetition of the whole term. In addition, we added the references of the test in the methodology part.

Comment 5

-Discussion please rewrite the first sentence of the first paragraph may be divided into two sentences.

 Respond Thanks for your kind suggestion. I have split the sentence into two sentences. Please see the discussion part.

Comment 6

-Since no raws were shown for review purposes, here is the sentence only to remove the term “always” and to delete it or insert the appropriate term.

“We noticed that the patients without positive family history had older age at diagnosis and longer diagnostic delay (more than 5 years) than patients with family history, which suggests the diagnosis of HD patients without family history is always delayed.”

 Respond Thanks for your kind suggestion. We have deleted the term “always” to make the sentence neutral.

 Comment 7

-Is this sentence finished? “Whereas previous studies reported that apathy increased with HD progression[20, 45].”

Respond We apology for the expression. We improved the sentence into “Whereas previous studies reported that apathy increased with HD progression20,48, which was inconsistent with our findings.”

Comment 8

-This sentence need to be rewritten “In summary, we first summarized the comprehensive clinical and genetic profile of Chinese HD patients.” “We first”?

 Respond Thanks for your suggestion. We deleted the word “first” to make the sentence better understood.

Comment 9

-The term “natural history” stands for?

 Respond The natural history of disease is the course a disease takes in individual people from its pathological onset ("inception") until its resolution (either through complete recovery or eventual death), which is from "Natural history of disease". A Dictionary of Epidemiology (ISBN 978-0-19-939005-2.). In order to express more appropriately, we rewrote the sentence that further longitudinal following up is needed to investigate the whole natural history of disease in China. Thanks for pointing out it.

Comment 10

-Please see some references related to the topic of your paper to include them if possible:

1

 Brain Sci. 2017 Jun; 7(6): 67.

Published online 2017 Jun 16. doi: 10.3390/brainsci7060067

 PMCID: PMC5483640

PMID: 28621715

 Neuropsychiatric Burden in Huntington’s Disease

Ricardo Augusto Paoli,1 Andrea Botturi,1,2,* Andrea Ciammola,3 Vincenzo Silani,3,4 Cecilia Prunas,1 Claudio Lucchiari,5 Elisa Zugno,1 and Elisabetta Caletti1

 2

A Systematic Review of Neuropsychiatric Symptoms and Functional Capacity in Huntington’s Disease

Jessie Sellers, M.S.N., Sheila H. Ridner, Ph.D., R.N., Daniel O. Claassen, M.D., M.S.

 Published Online:30 Aug 2019https://doi.org/10.1176/appi.neuropsych.18120319

 3

J Neuropsychiatry Clin Neurosci

 . 2022 Spring;34(2):100-112. doi: 10.1176/appi.neuropsych.21060154. Epub 2021 Dec 28.

Apathy and Huntington's Disease: A Literature Review Based on PRISMA

 Jihene Matmati 1, Christophe Verny 1, Philippe Allain 1

 PMID: 34961332 DOI: 10.1176/appi.neuropsych.21060154

 4

Archives of Psychiatric Nursing

Volume 35, Issue 3, June 2021, Pages 284-289

Archives of Psychiatric Nursing

Lifetime neuropsychiatric symptoms in Huntington's disease: Implications for psychiatric nursing

Author links open overlay panelJessie S.Gibsona12David A.Isaacsa2Daniel O.ClaassenaJeffrey G.Stovallb

https://doi.org/10.1016/j.apnu.2021.03.006

 5

J Neuropsychiatry Clin Neurosci

. 2022 Sep 21;appineuropsych21070191. doi: 10.1176/appi.neuropsych.21070191. Online ahead of print.

Apathy and Depression in Huntington's Disease: Distinct Longitudinal Trajectories and Clinical Correlates

Michael H Connors 1, Armando Teixeira-Pinto 1, Clement T Loy 1

PMID: 36128678 DOI: 10.1176/appi.neuropsych.21070191

 6

Australas Psychiatry

. 2018 Aug;26(4):366-375. doi: 10.1177/1039856218791036. Epub 2018 Jul 16.

Huntington's disease: Neuropsychiatric manifestations of Huntington's disease

Anita My Goh 1, Pierre Wibawa 2, Samantha M Loi 3, Mark Walterfang 4, Dennis Velakoulis 3, Jeffrey Cl Looi 5

 PMID: 30012004 DOI: 10.1177/1039856218791036

 Respond Thanks for your helpful suggestions. We have included four related and helpful references (PMID: 28621715, PMID: 31466515, PMID: 30012004, PMID: 30012004) in our paper.

Also, we have improved our English writing and expression, completed the methods description and clarified results presentation. We really appreciate the suggestions of yours and are willing to re-edit if needed.

Reviewer 2 Report

This manuscript is important for the patients and families impacted by HD in China.  Since China is a large country with may regions, the authors should state the catchment area and regions that the patients came from.  This information may impact the relevance related to other published studies and for generalizability.

The authors should state if the UHDRS was used with under license from the Huntington Study Group.

While the authors have nicely divided the population based on age, the age of determination should be explicitly stated.  It seems juvenile HD cut-off is 20, but "elderly-onset" is not clear.  Similarly, the authors should define stages.

The authors should consider speculating in the discussion section why the TFC is higher in the juvenile population.

Author Response

Response to Reviewer 2 Comments

Comment 1

-This manuscript is important for the patients and families impacted by HD in China.  Since China is a large country with many regions, the authors should state the catchment area and regions that the patients came from.  This information may impact the relevance related to other published studies and for generalizability.

 Respond Thanks for your helpful suggestions. We have stated the main region where the patients came from. Please see the results and discussion part. We also added the limitation that patients mainly came from southwestern China might cause population bias. Therefore, a multicenter study is needed to validate our findings.

 Comment 2

-The authors should state if the UHDRS was used with under license from the Huntington Study Group.

 Respond Thanks for your kind suggestion. We have added the statement in the methodology part.

Comment 3

- While the authors have nicely divided the population based on age, the age of determination should be explicitly stated.  It seems juvenile HD cut-off is 20, but "elderly-onset" is not clear.  Similarly, the authors should define stages.

 Respond Thanks so much for pointing out these unclear definitions. We have added the definition of elderly-onset HD patients and defined the disease stages according to the TFC score in the methodology part.

 Comment 4

-The authors should consider speculating in the discussion section why the TFC is higher in the juvenile population.

 Respond Thanks for your helpful suggestion. In fact, we found there were no differences in TFC scores among juvenile, adult and elder-onset HD patients. The juvenile population got high independence score (UHDRS-III) than the other two groups, which suggested juvenile HD might need less help from caregivers due to less chorea. However, the sample size of juvenile HD was small in our cohort, more patients are needed to get a more persuasive result.

Also, we have improved our English writing and expression, completed the methods description.

We really appreciate the suggestions of yours and are willing to re-edit if needed.

Reviewer 3 Report

1. The type of the manuscript should be described in the title.

2. IRB number should be provided

3. Provide a complete description of every abbreviation at first appearance. E.g., SPSS.

4. How were variables distributed?

5. Why did the authors not include individuals from other centers in China in the study?

6. Could the authors provide a table about different studies and their results in the Chinese population with HD?

Author Response

Response to Reviewer 3 Comments

Comment 1

  1. The type of the manuscript should be described in the title.

 Respond Thanks for your comment. We have added the research type in the title: The comprehensive analysis of motor and neuropsychiatric symptoms in patients with Hunting-ton’s disease from China: a cross sectional study.

Comment 2

  1. IRB number should be provided.

 Respond Thanks for pointing it out. We have added the IRB number (approval number 2015-236) in the methodology part.

Comment 3

  1. Provide a complete description of every abbreviation at first appearance. E.g., SPSS.

 Respond Thanks for pointing it out. We completed the detail description of SPSS and doublechecked the abbreviation to make sure the full name at the first appearance.

Comment 4

  1. How were variables distributed?

 Respond Thanks for your question. All the variables were normal distribution or approximately normal distribution which tested by Kolmogorov-Smirnov test and Shapiro-Wilk test. We added this point in the statistical analysis.

Comment 5

  1. Why did the authors not include individuals from other centers in China in the study?

 Respond Thanks for your good question. We do agree that recruitment of patients from other centers in China in the study would get the more persuasive result, while the study design of the paper is single center study. This limitation has been clarified in Discussion/Limitation.

Comment 6

  1. Could the authors provide a table about different studies and their results in the Chinese population with HD?

 Respond Thanks for the helpful suggestion. We have provided a table about comparison of two studies with large sample size of HD patients in China. Please see the supplementary table 3. The Li HL’s study had similar demographic and genetic features to our study, while we made more comprehensive analyses of clinical characteristics including motor and neuropsychiatric symptoms.

Also, we have improved our English writing and expression, completed the methods description and research design, cited more references relevant to the research.

We really appreciate the suggestions of yours and are willing to re-edit if needed.

Round 2

Reviewer 1 Report

Thank you for the revision according to the suggestions. Only please check and correct the references style in the manuscript text it should be [], and in the reference list, I believe there are minor errors to meet the JCM guidelines.

Author Response

Thanks for your helpful suggestions. We have checked and corrected the references style in the manuscript text and change it to be []. Also in the reference list, we have corrected the style of reference to meet the JCM guidelines. Please see the latest version of manuscript.